# Antimicrobial Activity of Bovine Bone Scaffolds Impregnated with Silver Nanoparticles on New Delhi Metallo-β-Lactamase-Producing Gram-Negative Bacilli Biofilms

Geiziane Aparecida Gonçalves [1], Victoria Stadler Tasca Ribeiro [1], Leticia Ramos Dantas [1], Ana Paula de Andrade [1], Paula Hansen Suss [1], Maria Alice Witt [2] and Felipe Francisco Tuon [1,*]

[1] Laboratory of Emerging Infectious Diseases, Graduate Program of Health Sciences, School of Medicine, Pontifícia Universidade Católica do Paraná, Curitiba 80215-901, PR, Brazil; geize35@gmail.com (G.A.G.); vicstadler@gmail.com (V.S.T.R.); leticia.dantas@pucpr.br (L.R.D.); ana_paula_de_andrade@hotmail.com (A.P.d.A.); paula.h@pucpr.br (P.H.S.)

[2] Chemistry Department, Pontifícia Universidade Católica do Paraná, Curitiba 80215-901, PR, Brazil; maria.witt@pucpr.br

* Correspondence: felipe.tuon@pucpr.br

**Abstract:** Introduction: The antibiofilm activity of silver nanoparticles has been extensively investigated in common bacteria. Metallo-β-lactamase-producing Gram-negative bacteria are hard-to-treat microorganisms with few therapeutic options, and silver nanoparticles were not evaluated on the biofilm of these bacteria. Objectives: The aim of this study was to evaluate the antibiofilm activity of a bone scaffold impregnated with silver nanoparticles in NDM-producing Gram-negative bacilli. Methods: Bone scaffolds from bovine femur were used for the tests and impregnated with silver nanoparticles (50 nm) by physical adsorption. Silver nitrate minimal inhibitory and bactericidal concentrations (MIC and MBC) were performed on NDM-producing *Escherichia coli*, *Klebsiella pneumoniae*, and *Pseudomonas aeruginosa*. Disc diffusion tests for silver nanoparticles' susceptibility and the quantification of biofilm production on plate and bone with sessile cell count were performed. Results: The MIC results demonstrated that silver nitrate had an antimicrobial effect on all microorganisms, inactivating the growth of isolates from a concentration of 8 μg/mL. MBC results showed that *E. coli* 16.211 was the only isolate to present MIC that were different from MBC, with a value of 16 μg/mL. Conclusion: Bone scaffolds impregnated with silver nanoparticles can significantly reduce the biofilm of multidrug-resistant bacteria. This is a strategical material that can be used as bone implant in different clinical conditions.

**Keywords:** biofilm; silver; bone; nanotechnology; impregnation; bacteria; NDM; scaffold

## 1. Introduction

The use of bone grafts is important in the reconstructive medicine of trauma and chronic bone diseases. Heterologous grafting is an option that has been widely used in the most varied areas of medicine. Its main representative is the bovine graft, which is easy to obtain, has great availability and is like human bone [1,2]. However, it may present limitations related to antigenicity [3]. With the aim of reducing antigenicity and preserving the inorganic matrix, the bovine graft undergoes washing processes, decellularization, is degreased and is subsequently dehydrated. Bovine bone has very similar chemical (composition), physical (porosity and size) characteristics and biological behavior to human bone, which favors osteoconduction. In addition, its inorganic matrix provides calcium and phosphorus content, which are essential to the formation of new bone tissue at the implant site [1,4].

The reconstruction of bone defects is still a considerable clinical challenge since several factors can hinder the success of the procedure. It can be associated with different complications, including infection of the graft with reabsorption of the bone tissue. These infections can be caused by local microbiota or even hospital-acquired pathogens, and biofilm is the cornerstone of infection maintenance [5]. The infections, especially those caused by multidrug-resistant bacteria and bacterial biofilms, are described with great relevance, as they are considered the most devastating complications of surgeries related to grafts, leading to long hospital stays, complex and prolonged treatments with local and systemic antimicrobials, and new surgery at the grafting site for debridement with possible loss of the graft and adjacent tissues [6–8]. Among the multi-drug resistant microorganisms, the emergence of New Delhi Metallo-β-lactamase (NDM1 carbapenemase) has presented a global public health problem. The high hydrolysis capacity of this β-lactamase, including against carbapenems, and their association with other plasmid-borne resistance mechanisms decreases the available therapeutic options [9]. In some cases, only polymyxin and cefiderocol are active drugs against these pathogens [10,11]. The bone infections associated with these pathogens have a disastrous evolution due to the few available antibiotics' lack of biofilm penetration.

Several antibiotics and other substances (e.g., bioglass and metals) have been used to impregnate the implant surface or even the bone graft. The antimicrobial activity of silver nanoparticles (AgNPs) has been extensively investigated, including through incorporating this metal into dental and orthopedic materials [12]. The size of the AgNPs increases their contact area with the bacterial cell membrane and causes permeability and easy penetration. Silver ions can interfere with the mitochondrial respiratory chair, causing oxidative reduction. Another supposed mechanism is the osmotic imbalance caused by the difference between extracellular and intracellular environments [13]. The impregnation of bone scaffolds can decrease trans- and postoperative infection rates with a high risk of failure, loss of bone volume, and chronic infectious processes, such as osteomyelitis.

There is a recent report of an experimental study concluding the non-reaction of a foreign body or inflammatory process in a bone graft impregnated with AgNPs [14]. However, the impregnation of bone graft with AgNPs has been poorly investigated, and microbiological studies are lacking in multidrug-resistant bacteria. Even though AgNPs have already been studied in NDM-producing *Pseudomonas aeruginosa* isolates, only a solution of AgNPs was evaluated [15]. Furthermore, antibiofilm AgNPs activity was not evaluated in NDM-producing microorganisms. Considering the severity of bone infections, the absence of a substitute with antimicrobial activity, and the few therapeutic options against NDM, mainly with antibiofilm activity, the purpose of this investigation was to evaluate the in vitro antibiofilm activity of a bone scaffold impregnated with AgNPs in NDM-producing bacteria.

## 2. Materials and Methods

### 2.1. Bone Scaffolds' Processing, Nanoparticles' Synthesis, and AgNPs' Impregnation

This was an experimental study conducted with cancellous bovine bone scaffolds models impregnated with AgNPs (Figure 1). The tests followed a process that meant that this project did not involve the use of live animals, as bovine bones were obtained directly from a slaughterhouse. Cancellous bone discs from the bovine femur were used for the tests. The discs of 6 mm in diameter and 3 mm thick (0.084 mL) (retrieved from a sagittal cut in the bone using a table band saw, using 6 mm diamond drill and small bone cylinders) (Figure 2) were made. Then, using a $\frac{1}{2}$ inch chisel, a hammer and a plastic caliper, the discs were measured and cut and processed using a modified, previously described protocol [16]. The procedure used to AgNPs synthesis was adapted from Dong et al. [17], where 8.5 mg of silver nitrate was dissolved (AgNO$_3$) in 45 mL of deionized water, followed by the addition of 14.7 mg of trisodium citrate (Na$_3$C$_6$H$_5$O$_7$). To this solution, 5 mL of a glucose solution (200 mg in 10 mL of deionized water) were dripped and maintained under vigorous stirring at 60 °C. After one hour of stirring, the yellowish-colored solution containing the silver

nanoparticles, 50 nm in size, was confirmed with ultraviolet light spectra and scanning electronic microscopy (SEM) as previously described [18].

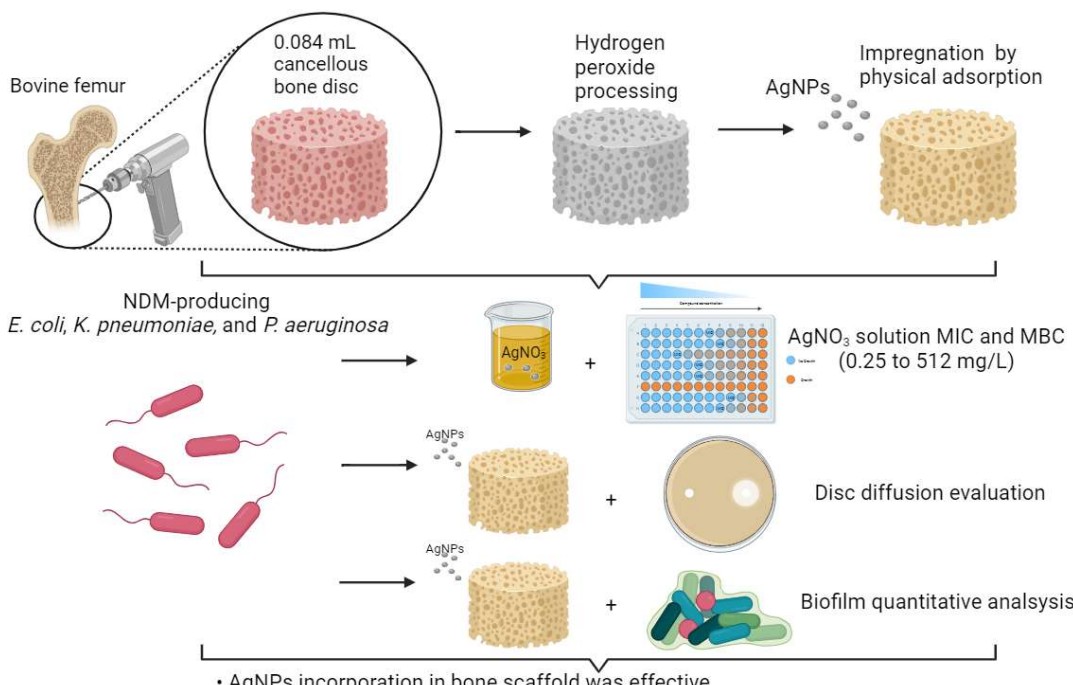

**Figure 1.** Method used for bone scaffold production and silver nanoparticles (AgNPs) adsorption, including microbiological tests.

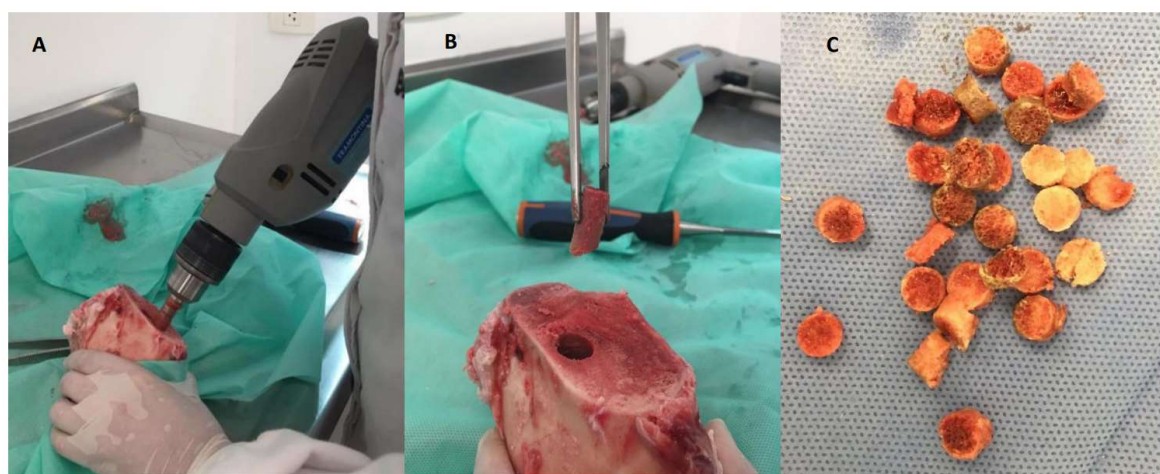

**Figure 2.** Bone scaffold processing. (**A**) 6 mm diamond drill punching the sagittal cut. (**B**) Displays the resulting bone cylinder after the drilling process. (**C**) Bone discs 6 mm in diameter and 3 mm thick.

Physical adsorption was used to incorporate the AgNPs in the bone matrix, based on an adaptation of the model reported by Becerril-Juárez et al. (2012) [19]. For this purpose, bone models were added to the AgNPs solution at room temperature for 60 min, and protected from light. After the incorporation of AgNPs in the bone matrices, the samples were kept at a temperature of 70 °C for 60 min in an oven with no air-circulation system for drying.

## 2.2. Qualitative Bone Disc Scaffold Analysis through Scanning Electronic Microscopy and Energy-Dispersion X-ray Spectroscopy

For the analysis of morphology and microstructural characteristics before and after AgNPs impregnation in the bone matrix, SEM was performed. The models were transferred to sterile glass Petri dishes with the primary fixative agent (0.68 g 99.5% sucrose, 0.42 g 98% sodium cacodylate, 0.6 mL 30% glutaraldehyde (Merck, Darmstadt, Germany) in 19.4 mL of deionized water, remaining in contact for 45 min. After contact with this primary fixative agent, the models were transferred every 10 min to the following solutions: buffer (composed of sucrose and cacodylate of sodium at the concentrations mentioned above), 35% ethanol, 50% ethanol, 70% ethanol, 100% ethanol and HMDS PA (hexamethyldisilazane) (Merck, Darmstadt, Germany). After fixation, the models were kept in a desiccator until the moment of visualization in SEM, when they were previously submitted to metallization with gold particles, in metallization equipment with a Q150R ES rotary pump (Quorum Technologies, Lewes, UK) and, later, fixed on a metallic base for observation in the scanning electron microscope JEOL JSM 6010PLUS-LA (JEOL, Tokyo, Japan) at an accelerating voltage of 20 kV. Observations were made at magnifications between 2000 and 100,000 times [18].

Energy Dispersion X-ray Spectroscopy (EDS) allows for the measurement of the chemical elements present in the sample as well as the determination of concentrations with great precision. The bone discs were characterized by EDS, before and after AgNPs impregnation in the bone matrix, in order to verify the presence and dispersion of silver on the bone tissue. EDS analysis was performed on the JEOL JSM 6010PLUS-LA Analysis Station with an accelerating voltage of 20 kV.

## 2.3. Microorganisms, Silver Nitrate Minimum Inhibitory Concentration (MIC) and Minimal Bactericidal Concentration (MBC), Disc-Diffusion Tests for AgNPs Susceptibility, and Biofilm Production and Evaluation on Bone

Three NDM-producing bacteria were included in this analysis, as follows: one *Klebsiella pneumoniae* 11.955, one *E. coli* 16.211, and one *P. aeruginosa* 20.589. The identification was confirmed by Matrix-Assisted Laser Desorption Ionization Time of Flight Mass Spectrometry (MALDI-TOF MS) (Bruker Daltonik, Bremen, Germany), and the *bla*NDM gene was identified and confirmed by whole-gene sequencing (MiSeq®, Illumina, San Diego, CA, USA).

There is no validation of the Minimal Inhibitory Concentration (MIC) for AgNP. Thus, silver nitrate MIC was performed as previously described [20], following the principles established by CLSI (Clinical Laboratory Standard Institute) [21]. A silver nitrate solution in progressive titers from 0.25 to 128 mg/L was used. a total of 200 μL aliquots of silver nitrate solution in Muller–Hinton broth (Oxoid Ltd., Basingstoke, Hampshire, UK) were placed, using different concentrations, in a 96-well plate, and 5 μL of the solution of each microorganism was inoculated to reach a concentration of 106 colony-forming units (CFU)/mL. After incubation for 24 h at 36 °C, the MIC was defined as the lowest concentration that did not show bacterial growth. All tests were performed in triplicate. In parallel, the minimal bactericidal concentration (MBC) for AgNPs was performed with the same microorganisms as presented above, plating, in triplicate, 100 μL of all solutions over the MIC. The MBC was determined as the final concentration without bacterial growth.

The microorganisms were previously diluted to a turbidity standard equivalent to 0.5 McFarland and inoculated on Mueller–Hinton agar plates (Laborclin—A Solabia Group, Pinhais, Brazil). In addition, two discs, one without impregnation (negative control) and one with AgNPs (test), were transferred to Muller–Hinton agar plates inoculated with different bacteria. The plates were incubated for 24 h at 36 °C. The analysis was quantitative, checking for the presence and measuring the diameter of the inhibition halo [22,23].

For biofilm production, we used the previously described protocols [24,25]. The experiment was executed in quintuplicate. From each microbe suspension, a 1:10 dilution was made in TSB (Tryptone Soy Broth) until concentrations of $10^7$ CFU/mL of bacteria were achieved. Then, 10 mL of TSB was poured into sterile 12-well plates until it completely

covered the bone discs (control, without AgNPs after processing, and impregnated AgNPs) for 2 h under agitation (120 rpm), so that cells could adequately adhere. The specimens were transferred to a new, sterile, 12-well plate containing 0.9% NaCl to remove planktonic cells from the material. Then, specimens were transferred to another sterile 12-well plate and submerged in 10 mL of TSB at 37 °C for 24 h without agitation. During this step, the cells that adhered to the device surface formed the biofilm. After this step, the specimens were submerged in 50 mL conical tubes filled with 10 mL of sterile 0.9% NaCl to remove the residues and unadhered/planktonic cells (step I). After this washing step, the specimens were allocated to 50 mL conical tubes filled with 10 mL of 0.9% NaCl for further processing (sonication), and the liquid of the last washing was stored for planktonic cells' analysis (step II).

Five specimens of each group were transferred to sterile conical tubes with 10 mL of 0.9% NaCl and sonicated for 15 min in an ultrasonic bath using a Soniclean 15 (Sanders Medical, Santa Rita do Sapucaí, Brazil) at a frequency of approximately 40 kHz and a temperature of 35 °C [26]. After the sonication step (step II), the supernatant (100 µL) was inoculated in TSA (Tryptone Soy Agar) for growth evaluation and cell counting (CFU/mL).

SEM and EDS data were descriptive and semi-quantitative, respectively. MIC and MBC were presented in µg/mL of silver nitrate. AgNPs activity data regarding agar diffusion were presented in millimeters (mm) and defined as the presence or absence of inhibition halo. To compare the cell count in the groups, the median CFU/mL obtained with quantitative culture was analyzed by the Mann–Whitney test and presented with a median with an interquartile range 25–75%. The difference in CFU/mL was significant when $p < 0.05$. The data were calculated, analyzed, and plotted using Prism 7.0 (Graphpad, San Diego, CA, USA).

## 3. Results

The procedure used to obtain AgNPs produced a yellowish color, observed with a maximum absorption peak around 420 nm, and measured by ultraviolet light spectra (Figure 3), suggesting the formation of AgNPs of about 50 nm in diameter. Nanoparticles' morphology and surface charge were characterized using the Malvern zetasizer. The size of AgNPs was found to be 51 ± 2.2 nm, as measured using Malvern Zetasizer (Figure 3 right). The surface charge of the particles was found to be −22.7 ± 5.5 mV, as measured by Malvern zetasizer. Using SEM and EDS, it is possible to observe the presence of nanoparticles impregnated in the bone scaffolds (Figure 4) due to the presence of two characteristic peaks in silver atoms, corresponding to 18.59% ($w/w$) of the sample. From EDS, one can confirm that the physical adsorption procedure used here was efficient in impregnating AgNPs and keeping them attached after drying the scaffolds (Figure 5).

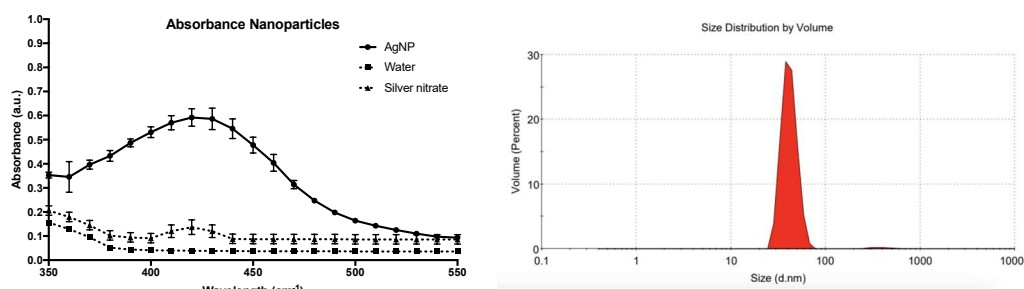

**Figure 3.** Absorbance curve of the nanoparticle, showing a peak at 420 nm, suggesting a size of 50 nm (**left**). The (**right**) image represents the peak of AgNPs by Malvern zetasizer, confirming the 50 nm size.

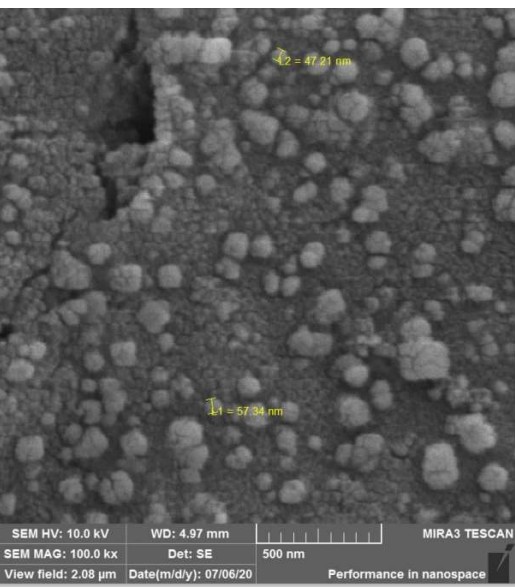

**Figure 4.** Scanning Electron Microscopy showing a morphological analysis of AgNPs on the bone surface at 100,000 times magnification. It was possible to visualize the AgNPs that were impregnated after treatment.

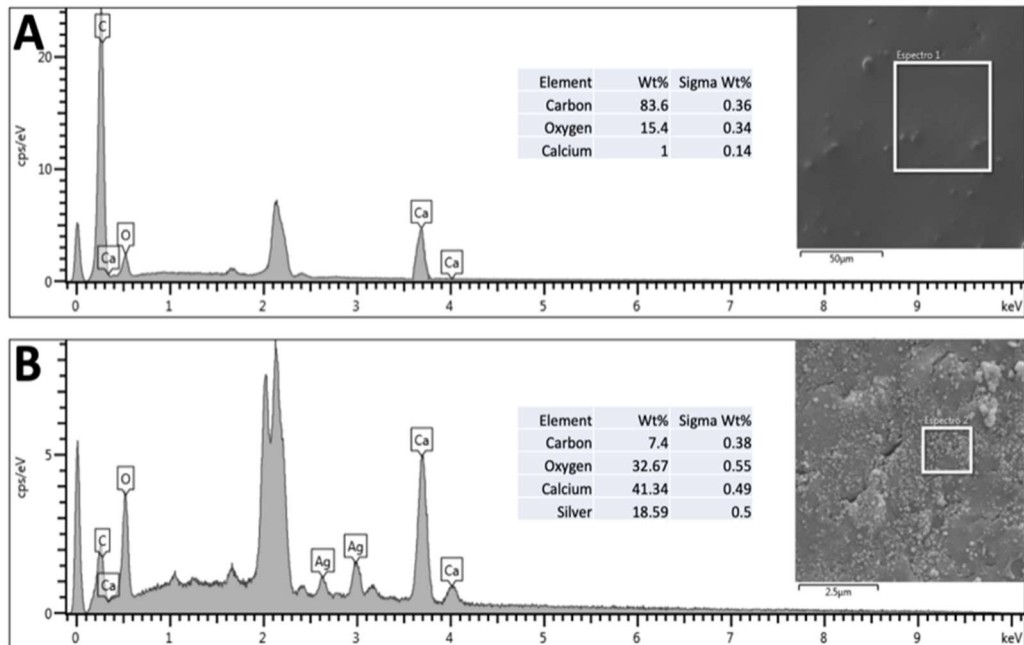

**Figure 5.** Graph resulting from the EDS analysis quantifying chemical components (C = carbon; O = oxygen; Ca = calcium; Ag = silver) of the model impregnated with AgNPs. (**A**) Control, before AgNPs' impregnation in the bone matrix. (**B**) Test, after AgNPs' impregnation in the bone matrix.

The MIC results demonstrated that silver nitrate had an antimicrobial effect on all microorganisms, inactivating the growth of isolates from a concentration of 8 μg/mL. MBC results showed that *E. coli* 16.211 was the only isolate to present an MIC that differed from that of MBC, with a value of 16 μg/mL.

It was possible to identify the presence of an inhibitory halo in the agar plates of all tested microorganisms. The *E. coli* 16.211 halo size was 15 mm, and in *P. aeruginosa* 20.589, it was 17 mm. The *K. pneumoniae* 11.955 plaque showed two inhibition halos, the internal one measuring 11 mm and the external one measuring 13 mm (Figure 6).

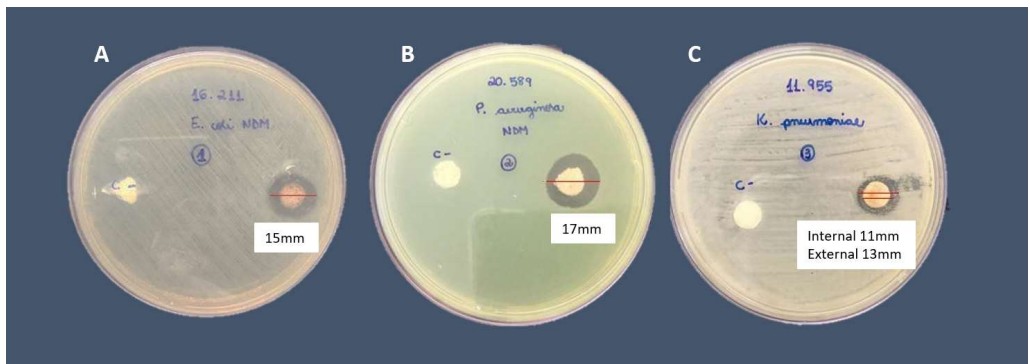

**Figure 6.** Antimicrobial activity of AgNP-impregnated bone scaffolds disc against (**A**) *E. coli* 16.211, (**B**) *P. aeruginosa* 20.589, and (**C**) *K. pneumoniae* 11.955.

AgNP-impregnated bone scaffolds showed a significant reduction in biofilm cell development (>3log CFU/mL) for *E. coli* ($p = 0.0078$) and *P. aeruginosa* ($p = 0.0079$). However, there was no significant decrease in *K. pneumoniae* ($p = 0.5476$) (Figure 7).

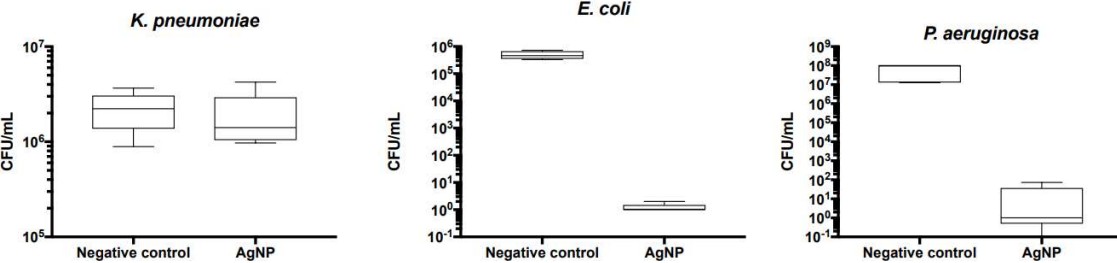

**Figure 7.** Cell count (CFU/mL) of biofilm of each evaluated microorganism in bone grafts with AgNPs and without impregnation (control).

## 4. Discussion

This study identified that three NDM-producing bacteria were susceptible to silver, but biofilm inhibition presented different results. Even so, it is important to identify that the bone scaffold impregnated with AgNPs may be an attractive graft against this type of microorganism, where therapeutic options may be null, and radical orthopedic procedures may be necessary, but with significant sequelae for patients, such as limb amputation [27]. Silver nanoparticles present an excellent anchoring mechanism with hydroxyl radicals from bone (hydroxyapatite) due to the negatively charged bone with a positive silver charge [28–30]. Several mechanisms are involved in the antibacterial potency of AgNPs. Due to their extraordinarily large surface area, AgNPs exhibit better antibacterial properties than metallic silver [31]. Even though all isolates were susceptible to silver, these values were obtained by silver nitrate, differently from other authors, who defined MIC by AgNPs, showing divergent results [32]. Mathur et al. described the antibacterial action of AgNPs through their penetration into the microorganism's cell membrane, increasing its permeability, leading to the rupture of this structure and consequent cell death [33]. Another mechanism that was described is the formation of free radicals that damage the cell membrane, causing porosity and cell lysis [34]. Mih et al. analyzed the mechanism of resistance to oxidative stress of the *E. coli* K-12 MG1655 strain, hypothetically punctuating the presence of amino acids capable of absorbing oxidative damage in the structural proteome of this bacterium [35].

Along with the bactericidal mechanism of silver nanoparticles, the relationship between particle size and antimicrobial effect has been reported [36]. Smaller particles have a higher surface area that allows for the release of a higher amount of silver ions, responsible for the most acceptable mechanism of bactericidal action. In this sense, smaller particles

are expected to present a greater antimicrobial effect than bigger particles and particle aggregates. This can explain the maintenance of viable bacterial cells after scaffold impregnation in the case of *K. pneumoniae*. Bacteria with a high capacity for multiplication and biofilm-producing, like *P. aeruginosa*, showed higher susceptibility to AgNPs. In the case of *K. pneumoniae*, we found a reduced activity of AgNPs on biofilm, which could be explained by the cationic characteristic of this biofilm [37]. Electrostatic interactions between negatively charged groups in the biofilm and the charged wire may also be expected to cause biofilm expansion when the wire is cathodic and contraction when the wire is anodic. A study that evaluated AgNPs with plant extracts demonstrated, in the evaluation of an isolated group of AgNPs, no change in the production of biofilms in KPC-producing *K. pneumoniae*, the same result as that found in our study [38].

Biofilm formation varies among different species of microorganism [39]. Olson et al. highlighted that *P. aeruginosa*, *E. coli*, and *Staphylococcus aureus* produce biofilms in a short period and without great nutritional requirements [40]. However, in our study, the broth used in the biofilm model presents abundant nutrients, which can cause differences in the biofilm pattern and bacterial redox system, affecting silver activity [41]. Anuj et al. evaluated the association of silver nanoparticles with linezolid as a bactericidal agent on *E. coli* MTCC® 443 ™, where the resistance mechanism of drug efflux of this microorganism was inactivated by AgNPs [42]. This is an important issue, considering that AgNPs are an important molecule for use in combination with antibiotics. The risk of using nanoparticles to treat biofilm lies in the fact that they activate factors that induce quorum sensing, which can increase biofilm production and also increase the population of sessile cells, especially at low concentrations of nanoparticles, inducing the production of virulence factors such as elastase, pyocyanin, and biofilms in *P. aeruginosa* [43]. On the other hand, high doses may have an inhibitory effect on quorum-sensing [44].

According to the plasmonic absorption spectrum of the AgNPs solution (Figure 3), it was possible to observe a peak at 420 nm, which shows the formation of nanometric silver, with an average size of around 50 nm, as compared to the study by Agnihotri et al. (2014) [45]. The average size of AgNPs is dependent on different factors, such as the type of reducing agent, their quantity and concentration, as well as the reaction temperature, agitation speed and duration of the reduction process [46]. The AgNPs synthesis mechanism proposed in this study involves the chemical reduction of silver cations ($Ag^+$) through the reducing agent (glucose solution) and subsequent nucleation and coalescence of metallic silver in particles. Then, trisodium citrate plays a key role as a stabilizer, preventing nanoparticles from aggregating [30,47]. The described procedure makes it possible to obtain a colloidal solution of AgNPs with a very stable yellow color. The stability of the system comes from the fact that the formed nanoparticles remain in suspension and do not aggregate to form larger particles. For this, trisodium citrate must be added more than to silver nitrate during synthesis, since this reagent is responsible for the stabilization of the formed nanoparticles. This occurs because citrate ions adsorb on the surfaces of nanoparticles, involving them and generating charges on these surfaces. These charges create an electrostatic repulsion between the nanoparticles, preventing their aggregation. For this reason, it is important to remember that reaction conditions, including stirring time and relative amounts of reagents, must be strictly controlled to obtain stable yellow colloidal silver. The concentration of trisodium citrate must be higher than that of silver nitrate, and if this ratio is modified, the aggregation of nanoparticles can occur in less than 1 h. The even greater addition of the stabilizing agent can increase the ionic strength of the medium and facilitate the aggregation of nanoparticles [30,47,48].

The physical adsorption of particles on surfaces, also called physisorption, involves a relatively weak interaction, which can be attributed to Van der Waals forces, and is a process that depends on several factors, such as surface area, and system temperature, as well as the nature of the adsorbate (liquid or gaseous substance that is retained on the surface), the nature of the adsorbent (solid substance that causes the retention of the substance), and the pH of the medium [49,50]. This type of adsorption results from the action of intermolecular

forces of attraction between the adsorbent and the adsorbed molecules, with no chemical bonds, and is therefore reversible. Furthermore, there is no change in the properties of the adsorbate or the adsorbent, which maintain their chemical natures [51,52]. In our study, in addition to being used as a stabilizer in the synthesis, trisodium citrate facilitated the process of physisorption of AgNPs on spongy bone tissue, since, through presenting carboxylate groups with negative charges, an interaction occurs through the intermolecular force of attraction with the positive charges of the bone mineral matrix, which is mainly composed of calcium ($Ca^{2+}$) [50,51]. In the graph shown in Figure 5B, it is possible to verify the presence of silver (Ag) peaks corresponding to 18.59% of the sample. From the EDS, it can be confirmed that the physical adsorption procedure that was used was efficient in impregnating the AgNPs and keeping them adsorbed after drying the models.

The MIC results demonstrated that silver nitrate had an antimicrobial effect on all microorganisms, inactivating the growth of isolates from a concentration of 8 µg/mL. These results were obtained using silver nitrate, unlike the results of John et al. (2020) [32], who defined the MIC for AgNPs, showing superior MIC results (12.5 µg/mL) for bacteria such as *E. coli*, *K. pneumoniae*, *Acinetobacter baumanii* and *S. aureus*, and also the study by Freire et al. (2018) [52], showing lower results of MICs for silver nitrate, (2.64 µg/mL–5.28 µg/mL) in *Aeromonas* spp. the MBC results showed that *E. coli* 16.211 was the only isolate to present an MIC different from MBC, with a value of 16 µg/mL, which may indicate resistance to the effect caused by AgNPs in vitro. Bacterial resistance to heavy metal ions may result from efflux pumps, and in Gram-negatives, they are prevented from reaching targets in the cell by the selectivity of the outer membrane [53–55]. While common antimicrobials generally act in the bacterial growth phase, such as in protein, nucleic acid and cell wall synthesis, an antibiofilm aims to achieve direct control of the biofilm, acting on the extracellular matrix of exopolysaccharides (EPS), destabilizing its structure, which is composed of calcium and magnesium ionic bridges, and inhibiting the formation and promoting the destruction of already consolidated biofilms [56,57]. Based on the reactivity of silver ions as an electron donor, it is believed that these ions have a considerable effect on the overall stability of the exopolysaccharide matrix [58]. There is still no evidence regarding the mechanisms of action of AgNPs in biofilms; however, it is suspected that their antibiofilm potential is directly related to their size, which may facilitate their penetration through the pores present in the exopolysaccharide layer [59–61].

In a biofilm model of *Pseudomonas putida*, developed by Fabrega and collaborators (2009) [62], AgNPs were able to cross the biofilm and reach the interior of bacterial cells, promoting a decrease in their cellular activity, and the reduction in the biofilm was identified by a process the authors called "biofilm desquamation". In another study by Kalishwaralal et al. (2010) [59], *P. aeruginosa* and *Staphylococcus epidermidis*' ability to synthesize EPS was blocked at low concentrations 50 nM (nanomolar) of silver nanoparticles. By increasing the concentration to 100 nM, AgNPs not only effectively inhibited bacterial growth, but also interfered with the already consolidated biofilm. Considering the results of our study, this bone framework has the potential to decrease the biofilm synthesis of *E. coli* and *P. aeruginosa* by decreasing the number of viable sessile cells. However, the same did not occur for the *K. pneumoniae* strain. In the same way that a second halo was observed in the agar in the analysis of the antimicrobial activity of AgNPs impregnated in bone discs, demonstrating a possible heteroresistance, it is expected that this strain would also be more resistant in the quantitative analysis of sessile cells. The identification of a heteroresistant strain is very important to define the choice of treatment, since the use of AgNPs in these cases may not be completely feasible. This study has some limitations because we evaluated few bacteria, and in vitro model cannot represent a real-life biofilm in osteomyelitis. Even though the literature on impregnated materials is borad, microbiological studies with biofilm on bovine scaffolds have not been reported to date. Moreover, new tests must be carried out to improve the action of silver nanoparticles on other microorganisms, enabling a greater understanding of the antimicrobial potential presented by these particles.

## 5. Conclusions

Considering the current results of our study, this bone scaffold has both antimicrobial and anti-biofilm properties against NDM-producing bacteria. AgNP incorporation is effective, decreasing biofilm synthesis by reducing the viability of sessile cells, suggesting that it a promising material for clinical use in dental and orthopedic procedures. The AgNPs graft could be used as a material for infection prevention, as well as a graft for the treatment of infections by multidrug-resistant bacteria.

**Author Contributions:** G.A.G.—conceptualization, microbiological studies, writing of the draft; V.S.T.R.—microbiological studies, manuscript writing and review; L.R.D.—microbiological studies, results' analysis; A.P.d.A.—microbiological studies, results' analysis; P.H.S.—microbiological studies, results' analysis; M.A.W.—nanoparticles' processing and impregnation; F.F.T.—manuscript writing and review, data analysis. All authors have read and agreed to the published version of the manuscript.

**Funding:** This research received no external funding.

**Institutional Review Board Statement:** Ethical review and approval were waived for this study, due to the bones under investigation being from bovine that had already been slaughtered within the commercial value chain.

**Informed Consent Statement:** Not applicable.

**Data Availability Statement:** The data that support the findings of this study are available from the corresponding author upon reasonable request.

**Acknowledgments:** We thank Francisco Carlos Serbena and technician Vanessa with the FEG in the C-LABMU from Universidade Estadual de Ponta Grossa. We thank Laborclin for the culture media donation.

**Conflicts of Interest:** F.F.T. is a CNPq researcher. The other authors declare no conflict of interest.

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
