# Peer review of "Antimicrobial Activity of Bovine Bone Scaffolds Impregnated with Silver Nanoparticles on New Delhi Metallo-β-Lactamase-Producing Gram-Negative Bacilli Biofilms"

_compounds, doi:10.3390/compounds3040042_

Round 1

Reviewer 1 Report

Comments and Suggestions for Authors

The authors adequately demonstrate the preparation of the bone support, the synthesis of silver nanoparticles in the antibiofilm activity and the inhibitory effect on three types of microorganisms: E. coli, P. aeruginosa and K. pneumoniae.

In this sense, I consider it important to provide a more in-depth explanation of why silver nanoparticles do not have a similar effect on the K. pneumoniae microorganism compared to the other two.

On the other hand, the anchoring mechanism of the metal nanoparticles to the structure of the hydroxyhapatite is mainly due to the interactions of the hydroxyl groups of this (negatively charged) and the charge that tends to positive of the silver nanoparticles, there are reports of I study these interactions using XPS

Author Response

The authors adequately demonstrate the preparation of the bone support, the synthesis of silver nanoparticles in the antibiofilm activity and the inhibitory effect on three types of microorganisms: E. coli, P. aeruginosa and K. pneumoniae.

In this sense, I consider it important to provide a more in-depth explanation of why silver nanoparticles do not have a similar effect on the K. pneumoniae microorganism compared to the other two.

Answer: Perfect, we have included a deeper discussion about this issue

On the other hand, the anchoring mechanism of the metal nanoparticles to the structure of the hydroxyhapatite is mainly due to the interactions of the hydroxyl groups of this (negatively charged) and the charge that tends to positive of the silver nanoparticles, there are reports of I study these interactions using XPS

Answer: thank you very much. We have included some publications and improved the discussion with this comment.

Reviewer 2 Report

Comments and Suggestions for Authors

The manuscript entitled “Antimicrobial activity of bovine bone scaffolds impregnated with silver nanoparticles on New-Delhi-Metallo-β-lactamase producing Enterobacterales biofilms” written by Gonçalves and coworkers presents a study detailing preparation and antibacterial assessment of bovine bone scaffolds constructs enriched with silver nanoparticles. The research is interesting, as the particles were successfully obtained in the nano size of about 50 nm in diameter and demonstrated noticeable activity. Moreover, adsorption ratio of these particles onto the constructs was found to be reasonable. I support the paper for its publication in Compounds with only minor revisions required:

1. The introduction lacks clarity and cohesion between paragraphs. It is advisable to emphasize the primary objective of the study for better context.

2. I appreciate the comprehensive nature of the discussion. However, it would be valuable if the authors could include information from existing literature or research regarding the impact of silver nanoparticles on quorum sensing within biofilms. Integrating this aspect into the discussion would enhance the paper's overall value.

Author Response

The manuscript entitled “Antimicrobial activity of bovine bone scaffolds impregnated with silver nanoparticles on New-Delhi-Metallo-β-lactamase producing Enterobacterales biofilms” written by Gonçalves and coworkers presents a study detailing preparation and antibacterial assessment of bovine bone scaffolds constructs enriched with silver nanoparticles. The research is interesting, as the particles were successfully obtained in the nano size of about 50 nm in diameter and demonstrated noticeable activity. Moreover, adsorption ratio of these particles onto the constructs was found to be reasonable. I support the paper for its publication in Compounds with only minor revisions required:

  1. The introduction lacks clarity and cohesion between paragraphs. It is advisable to emphasize the primary objective of the study for better context.

Answer: Thank you. We have improved the text of introduction, including a better justification and objective.

  1. I appreciate the comprehensive nature of the discussion. However, it would be valuable if the authors could include information from existing literature or research regarding the impact of silver nanoparticles on quorum sensing within biofilms. Integrating this aspect into the discussion would enhance the paper's overall value.

Answer: Perfect. This is an important issue that was lacking in the discussion. We have included some data about it.

Reviewer 3 Report

Comments and Suggestions for Authors

Although the paper is significant, there are several issues many missing parts should be addressed before decision , that I recommend major revision

1. Please, check the errors/typos throughout manuscript

2. Article structure and references should be checked according to journal standards.

3. Qualify of images should be improve according to journal standards.

4 Introduction section lacks suitable references and information. Kindly add some more relevant information and cite articles in introduction section

5. As of now, the introduction section contains several small paragraphs. Combine some paragraphs and make just 3 paragraphs. 

6. Why authors have not analyzed the particle size using DLS. PDI needs to be determined for ensuring particle size homogeneity. 

7. Zeta potential of nanoparticles has to be determined.

8. Further characterization studies should be required such as  atomic absorption to confirm and determine the present of the prepared nanoparticles .

9. Presently, the results and discussion section is too clumsy. Rearrange small paragraphs for better readability.

10. Abstract and conclusion section should be modified to show importance of study

Comments on the Quality of English Language

 Extensive editing of English language required

Author Response

Although the paper is significant, there are several issues many missing parts should be addressed before decision , that I recommend major revision

  1. Please, check the errors/typos throughout manuscript

Answer: The text was entirely revised by an English proofreading company.

  1. Article structure and references should be checked according to journal standards.

Answer: Thank you. We have revised the journal standards and used the word template.

  1. Qualify of images should be improve according to journal standards.

Answer: Perfect. We have improved image quality.

4 Introduction section lacks suitable references and information. Kindly add some more relevant information and cite articles in introduction section

Answer: Thank you.  We have included more data and references in the introduction.

  1. As of now, the introduction section contains several small paragraphs. Combine some paragraphs and make just 3 paragraphs. 

Answer: Ok. Considering that more data were included, we combined some paragraphs with a total of 4 paragraphs.

  1. Why authors have not analyzed the particle size using DLS. PDI needs to be determined for ensuring particle size homogeneity. 7. Zeta potential of nanoparticles has to be determined. 8. Further characterization studies should be required such as atomic absorption to confirm and determine the present of the prepared nanoparticles .

Answer: Perfect. Nanoparticles morphology and surface charge were characterized using Malvern zetasizer.

  1. Presently, the results and discussion section is too clumsy. Rearrange small paragraphs for better readability.

 Answer: Thank you for this suggestion. We have improved the format of discussion.

  1. Abstract and conclusion section should be modified to show importance of study

Answer: We have changed the text to improve the importance.